# Role of Integrins in Modulating Smooth Muscle Cell Plasticity and Vascular Remodeling: From Expression to Therapeutic Implications

**DOI:** 10.3390/cells11040646

**Published:** 2022-02-13

**Authors:** Manish Jain, Anil K. Chauhan

**Affiliations:** 1Pharmacology Division, University Institute of Pharmaceutical Sciences (UIPS), Panjab University, Chandigarh 160014, India; 2Department of Internal Medicine, Division of Hematology/Oncology, University of Iowa, Iowa City, IA 52242, USA

**Keywords:** integrins, smooth muscle cell, phenotype switching, neointimal hyperplasia, restenosis, extracellular matrix, fibronectin

## Abstract

Smooth muscle cells (SMCs), present in the media layer of blood vessels, are crucial in maintaining vascular homeostasis. Upon vascular injury, SMCs show a high degree of plasticity, undergo a change from a “contractile” to a “synthetic” phenotype, and play an essential role in the pathophysiology of diseases including atherosclerosis and restenosis. Integrins are cell surface receptors, which are involved in cell-to-cell binding and cell-to-extracellular-matrix interactions. By binding to extracellular matrix components, integrins trigger intracellular signaling and regulate several of the SMC function, including proliferation, migration, and phenotypic switching. Although pharmacological approaches, including antibodies and synthetic peptides, have been effectively utilized to target integrins to limit atherosclerosis and restenosis, none has been commercialized yet. A clear understanding of how integrins modulate SMC biology is essential to facilitate the development of integrin-based interventions to combat atherosclerosis and restenosis. Herein, we highlight the importance of integrins in modulating functional properties of SMCs and their implications for vascular pathology.

## 1. Introduction

Vascular smooth muscle cells (SMCs), present in the media layer of arteries, are critical to maintain the vascular tone of resistance arteries through synergic action between vasodilators/vasoconstrictors and vascular SMC contractility. SMCs exist in a differentiated, contractile, non-proliferative state in healthy arteries and exhibit an elongated myocyte morphology. Differentiated SMCs are characterized by the expression of a repertoire of smooth muscle-specific contractile and cytoskeletal proteins (e.g., SM-myosin heavy chain (*MYH11*), smooth muscle alpha-actin (*ACTA2*), SM22α (*TAGLN*), calponin (*CNN1*), h-caldesmon (*CALD1*), and smoothelin (*SMTN*)), all of them required to maintain the integrity of the arterial wall [1]. Unlike other differentiated cells, SMCs are not terminally differentiated and have the flexibility to shift from a contractile to a proliferative, pro-migratory, synthetic phenotype, exhibiting a rhomboid morphology. These de-differentiated SMCs are characterized by reduced expression of contractile proteins [2]. The transition of SMCs from a “contractile” to a “synthetic” phenotype is known as SMC phenotypic modulation or switching, which contributes to SMC proliferation, and migration, and thereby plays a vital role in the progression of atherosclerosis, in-stent restenosis, and other cardiovascular hyperplastic disorders. Additionally, platelet-derived growth factor (PDGF), transforming growth factor-beta (TGF-β), cytokines, integrins, angiotensin II, nitric oxide, reactive oxygen species, and the components of the extracellular matrix (ECM) are known to modulate SMC phenotype [3,4,5,6]. These dedifferentiated SMCs re-enter the cell cycle and secrete ECM components, including fibronectin, which contributes to vascular remodeling. In fact, lineage-tracing experiments suggest that phenotypically modulated SMCs within lesions can comprise about ~30% of the total cell count [7]. In healthy arteries, SMCs are surrounded by ECM components including laminin, collagen type IV, and heparan sulfate proteoglycan. Following injury, SMC activation is associated with marked changes in ECM composition, such as the disappearance of laminin and other basement membrane structures, and the appearance of abundant deposits of fibronectin and vitronectin around proliferative cells in the media and intima [8,9], suggesting a functional role of ECM in SMC activation. Integrins are the primary ECM receptors that regulate cell–cell and cell–ECM interactions. Furthermore, the adhesion, proliferation, and migration of SMCs are regulated by the interaction of integrins with ECM components [10]. 

## 2. Integrins: A Brief Overview

Integrins are transmembrane heterodimeric receptors that bind to cytoskeletal proteins of SMCs, including talin, vinculin, α-actinin, and filamin, and play a key role in SMC biology and in the development, maintenance, and remodeling of the vasculature [11,12,13]. The integrin family includes 18 alpha (α) and 8 beta (β) subunits that form 24 distinct αβ heterodimers. Each integrin heterodimer consists of a large extracellular domain region, two single-pass transmembrane helices (one in each subunit), and short cytoplasmic tails [14,15]. Integrins are known to adopt three central conformational states: inactive (low affinity, predominant state), active (high affinity, intermediate state), and ligand occupied (active state). Integrins can transmit signals from inside the cell to outside (inside-out signaling) and from outside to inside the cell (outside-in signaling). The process involves intracellular binding of ligands to the cytoplasmic domain, which causes a major change in the extracellular domain of the integrin receptor, leading to a high affinity for extracellular ligands [16,17]. Integrin outside-in signaling regulates cell growth, cell survival, and SMC-ECM interaction [2,16]. The activation of cell surface receptors, including growth factor receptors and cytokine receptors, also results in some conformational change in integrin receptors that, in turn, modulates its ligand-binding characteristics.

Depending on their ligand recognition pattern, integrins are classified as laminin-binding integrins (α3β1, α6β1, α7β1, and α6β4), collagen-binding integrins (α1β1, α2β1, α10β1, and α11β1), leukocyte-binding integrins, and Arg-Gly-Asp (RGD) binding integrins [14]. Laminin-binding integrins mediate the adhesion of cells to basement membranes; collagen-binding integrins mediate the adhesion of cells to collagen and chondroadherin; leucocyte-binding integrins bind intercellular adhesion molecule (ICAM) and plasma proteins. In contrast, RGD-binding integrins recognize three amino acid motifs, the ‘arginine-glycine-aspartic acid’ sequence commonly found in several ECM components, including vitronectin, fibronectin, fibrinogen, and von Willebrand factor [14]. Among the 24 human integrin subtypes known to date, eight integrin dimers recognize the tripeptide RGD motif within ECM proteins, namely: αvβ1, αvβ3, αvβ5, αvβ6, αvβ8, α5β1, α8β1, and αIIbβ3. 

## 3. Role of Integrins in SMC Biology

Integrin signaling plays an essential role in SMC biology by regulating adhesion, migration, proliferation, contraction, and differentiation [10,16,18,19,20,21]. Several proteins such as integrin-associated protein, integrin-linked kinase, focal adhesion kinase (FAK), tetraspanin CD9, and urokinase-type plasminogen activator receptor modulate integrin-mediated cell motility and adhesion in SMCs [10,22,23,24,25]. Integrin signaling in SMC also involves growth factor receptors that crosstalk between signaling pathways [17,26]. Several studies suggest that synergism may occur between integrin and downstream signaling molecules [17,27]. For example, integrin-mediated adhesion to ECM can enhance growth factor signaling on its receptor. In some cases, interactions with ECM may aid in the effective presentation of the growth factors to their receptors [28]. Additionally, integrin activation includes receptor tyrosine phosphorylation [29]. For instance, integrin–ligand adhesion triggers FAK auto-phosphorylation at tyrosine (Tyr) 397, which prompts FAK association with steroid receptor coactivator (Src). Src then phosphorylates other tyrosine residues that contribute to the full activation of FAK [25]. The activated FAK/Src complex facilitates various key signaling cascades, including the activation of serine-threonine protein kinase (AKT), extracellular signal-regulated kinase (ERK), and p38 mitogen-activated protein kinase (MAPK) pathway [30,31], all of which are known to regulate SMC proliferation and migration. A schematic summary of the proposed mechanism is shown (Figure 1).

Integrin–ligand interactions play a crucial role in remodeling of the injured vessel wall during wound healing, arterial stent injury, and in maintaining typical vascular structure [32]. Several integrins contribute to SMC activation. The major α-integrin subunits present in SMC are α1, α3, α5, α8, and α9 [10,33], whereas β subunits are β1, β3, and β5. The expression of integrins is dynamic and varies dramatically in SMC with different phenotypes [21,33]. Few integrins are upregulated in activated SMC, while expression levels are very low or undetectable in differentiated quiescent SMCs [21]. For example, integrin α1β1 is a collagen-binding integrin that is highly expressed in resting SMCs, and its expression is significantly downregulated in culture conditions [34]. Similarly, integrin α8β1 is overexpressed in SMCs that display a contractile phenotype, and its expression is downregulated after vascular injury [35]. Studies have demonstrated that the downregulation of integrin α8β1 causes actin filaments (a hallmark feature of contractile SMC phenotype) to dissociate and subsequently disintegrate, favoring a synthetic SMC phenotype [12]. Other integrins, including α2β1, α5β1, α5β3, and α4β1, are often expressed on the surface of SMCs in a low-affinity ligand-binding conformation [18,19,36,37,38,39]. The α5β1, which is a receptor for fibronectin, is poorly expressed in quiescent vessels in vivo. Following injury, fibronectin and integrin α5β1 expression is upregulated [18]. Another integrin subunit β3 is also known to be upregulated in response to stimuli, such as mechanical injury and neointimal hyperplasia, whereas blocking β3 attenuates SMC migration [40]. Several other integrins, including α2β1, α5β1, α5β3, and α4β1, are known to contribute to SMC migration and synthetic phenotype [38,41], whereas α1β1 [42] and α7β1 [20] were shown to mediate the phenotypic switch of SMCs. 

## 4. Role of Integrins in Neointimal Hyperplasia

Neointimal hyperplasia refers to post-intervention, pathological vascular remodeling due to the proliferation and migration of SMCs into the intimal layer, resulting in vascular wall thickening. During neointimal recruitment, SMCs are exposed to various ECM proteins, and integrin-ECM signaling has been shown to drive smooth muscle fibroproliferative remodeling. Several integrins are also known to promote neointimal hyperplasia, and evidence suggests that blocking integrins such as αIIbβ3 [43] and α4β1 [44,45] prevents neointimal hyperplasia. Besides these, the current literature strongly supports a role of signaling through αvβ3 in SMCs during neointimal hyperplasia [32]. In humans, αvβ3 is present in normal arteries and at the sites of SMC accumulation in atherosclerotic plaques. Several studies have shown that targeting αvβ3 integrin limits neointimal hyperplasia in small animal models of restenosis, including rat, rabbit, hamster, and guinea pig carotid angioplasty models [32,40,46,47]. In addition, an antibody to β3 integrin was demonstrated to prevent the development of intimal hyperplasia in wild-type diabetic mice [48]. Although β_3_-integrin blockade effectively reduces neointimal hyperplasia in animal models, the genetic ablation of β3 was found not to be effective for preventing intimal hyperplasia in animal models [49]. Therefore, it was speculated that the genetic loss of β3 might result in compensatory increases in the number and affinity of other adhesion receptors. In contrast, such compensation probably cannot occur with acute inhibition of αvβ3. In addition, the absence of β3 may affect signaling mediated by other integrins by decreased binding of intracellular proteins involved in signaling that ordinarily bind to the cytoplasmic domain of the missing integrin. Besides its detrimental role, some integrins are also known to prevent neointimal hyperplasia, such as α8β1 [50] and α7β1 [51]. The expression of different integrins on SMC, their ECM ligand, and their possible role in SMC function and neointimal hyperplasia are summarized in Table 1.

## 5. Integrin α9- An Overlooked Integrin

Several ECM proteins, which are generally expressed at low levels in normal adult tissues, are highly expressed during vascular remodeling [36,70]. Examples of such proteins include OPN, Tenascin-C, and cellular fibronectin containing EDA (Fn-EDA), all of which are known to promote SMC proliferation and neointimal hyperplasia [71,72,73]. It is important to note that Fn-EDA contains a non-RGD sequence known to interact with integrin α9 [74]. Furthermore, α9 and its matrix protein ligands associate with and synergize signaling from several growth factors, including PDGF-BB, to promote cell adhesion and motility [31,75,76]. In addition, α9 is known to be expressed by SMC [77], suggesting an essential role of α9 in SMC biology. In humans, α9 is encoded by the ITGA9 gene, located in the 3p21.3–22.2 segment of a chromosome, which encodes the polypeptides of 1035 amino acids and has a size of 114.5 KD [78]. The structure of α9 consists of a ligand-binding large N-terminal extracellular domain, a transmembrane segment, and a short C-terminal cytoplasmic domain that specifically binds to intracellular proteins [78]. α9 exclusively heterodimerize with β1 subunit, generating α9β1 heterodimer. Unlike other integrins that recognize RGD sequence, α9β1 recognizes a Met-Leu-Asp sequence, and it forms a unique subfamily with α4β1. Initially, due to many shared ligands, α9 was thought to have similar functions to those of α4; however, genetic ablation studies in mice revealed that phenotypes do not overlap, suggesting different functions in vivo [79]. Besides SMCs, integrin α9 is widely distributed throughout airway epithelium, skeletal muscle, endothelial cells, smooth muscle, hepatocytes, neutrophils, and cancer cells and has been shown to have an important role in regulating cell adhesion, migration, wound healing, thrombosis, angiogenesis, and inflammatory and immune responses [80,81,82,83]. 

## 6. Role of α9β1 in SMC Proliferation and Neointimal Hyperplasia

In recent years, α9β1 has gained particular attention because of its involvement in many diseases, including rheumatoid arthritis and multiple sclerosis [84]. In quiescent murine and human aortic SMCs, α9 is expressed at low levels and is mainly restricted to a membrane lining. After stimulation with PDGF-BB, which is known to modulate the membrane mobility and trafficking of integrins [85], a higher expression of α9 was detected in the cytoplasm (Figure 2) [21]. SMC-specific deletion of α9 significantly reduces SMC proliferation, migration, phenotypic switching, and injury-mediated pathological remodeling [21]. Previously, it was demonstrated that integrin and growth factor receptors activate the GSK3β signaling pathway [86]. GSK3β is known to phosphorylate β-catenin, the central signaling molecule of the canonical Wnt pathway, making it available for proteasomal degradation [87]. Nuclear-localized β-catenin interacts with TCF/LEF family of transcription factors and promotes its target gene expression. α9 deficiency was associated with higher GSK3β activity [86]. Studies suggest that integrin α9β1 utilizes the common integrin signaling proteins, including FAK, Src, and ERK [30,31]. Utilizing human coronary and mouse aortic SMCs, SMC-specific α9-deficient mice, and blocking antibody to the α9 subunit, we demonstrated that α9 activates FAK, Src, ERK, and p38 pathway and regulates nuclear translocation of β-catenin. Although the precise molecular mechanism of α9β1-induced FAK-Src activation remains to be elucidated, we speculate that Src may directly interact with the cytoplasmic tail of α9. Recently, Kurotaki et al. developed anti-integrin α9 antibody (clone 55A2C), which was shown to have an inhibitory effect on the binding of α9/NIH3T3 cells to the synthetic peptides AEIDGIEL, a sequence similar to the EDGIHEL sequence present in the EDA segment of fibronectin [88]. 55A2C has been shown to have an inhibitory effect on arthritis and multiple sclerosis progression in murine models [84]. We demonstrated that pretreatment with 55A2C suppressed PDGF-induced SMC proliferation and migration and inhibited injury-induced neointimal hyperplasia [21]. 

## 7. Role of Cellular Fibronectin Containing EDA Domain in α9 Mediated SMC Activation

The predominant isoforms of fibronectin found in the ECM, known as cellular fibronectin (cFn), are dimeric or cross-linked multimeric structures containing either alternatively spliced extra domain A (EDA) or extra domain B (EDB) or both, in varying proportions [89,90]. cFn containing EDA (Fn-EDA) in the ECM is synthesized by vascular cells, including endothelial cells, and its expression levels are upregulated during the development of neointima [71,91]. Recently, we demonstrated that PDGF-BB upregulates cellular Fn-EDA in stimulated SMCs, promotes phenotype switching and proliferation via TLR4, and promotes neointimal hyperplasia [71]. Furthermore, using RGDS peptide, we found that Fn-EDA mediates SMC proliferation and migration partially through integrins that are not recognized by RGDS peptide [71]. Notably, α9β1 regulates the functional activity of SMC through a variety of several non-RGD sequences such as SVVYGLR in OPN [92], AEIDGIEL in Tenascin-C [93], and PEDGIHELFP in cellular Fn containing EDA [88]. In line with these observations, studies using α9-deficient SMCs and recombinant EDA-containing or EDA-lacking peptides found that integrin α9 mediates SMC proliferation, migration, and phenotypic switching partially via Fn-EDA. These studies unequivocally support a causal connection between integrin α9 and FN-EDA in SMC proliferation and neointimal hyperplasia exacerbation.

## 8. Anti-Integrin Therapies in SMC Proliferation and Injury-Induced Neointimal Hyperplasia

Many studies have focused on targeting integrins as an intervention for aberrant SMC proliferation (Table 1). Studies with α5β1- and αvβ3-specific antagonists, RGD peptide, or integrin blocking antibody in a variety of preclinical models demonstrated that targeting of these integrins inhibit proliferation and migration of SMC and prevented neointimal hyperplasia [33,94,95]. Early clinical trials found that Abciximab/c7E3 (a mouse/human chimeric Fab portion of the IgG), integrilin/eptifibatide (an RGD based inhibitor), and AGGRASTAT/Tirofiban (a non-peptide tyrosine derivative) improved early adverse cardiac events after percutaneous coronary interventions (PCI) [47,96,97,98,99]. However, subsequent clinical trials did not meet expectations and demonstrated that these antagonists did not prevent intimal hyperplasia [47,100,101]. It is important to note that in these clinical trials, integrin antagonists were infused for the short term that primarily inhibited platelet activation. There is a possibility that such a short-term infusion could not block αvβ3 during vascular restenosis. Therefore, clinical trials specifically designed to assess the long-term effect of integrin-blocking on clinical restenosis are required. Another study tested the safety and efficacy of abciximab-coated stents in native human coronary artery lesions [102]. The 6-month intravascular ultrasound analysis showed that the area of neointimal hyperplasia was significantly smaller in the abciximab-coated stent group than in the control stent group, which indicated abciximab-coated stents to be safe and effective in the prevention of coronary restenosis [103]. Unfortunately, in a 2-year follow-up, abciximab-coated stents did not show superior clinical outcomes over bare-metal stents. Further studies are warranted to confirm these results in large-scale, prospective randomized trials.

Although the results of initial clinical trials with some integrins were disappointing, a study found that stents coated with peptide targeting αvβ3 decreased neointimal growth and improved vessel healing and reendothelialization in iliac arteries of New Zealand white rabbits [46]. Other preclinical studies demonstrated that targeting integrins with RGD peptides results in reduced neointimal growth [40,104]. Still, the clinical development of RGD-based integrin inhibitors has faced significant challenges, as many of these linear peptides have bioavailability and selectivity issues. In addition, many non-RGD recognizing integrin heterodimers contribute to SMC biology. Examples of such integrins include α4β1, α4β7, and α9β1. Blocking VCAM-1 and α4β1 interaction using anti-α4 integrin antibody or α4 integrin inhibitor (ELN 457946) was shown to attenuate neointimal formation [105] and in-stent restenosis [45].

## 9. Clinical Perspective

Balloon angioplasty followed by stent implantation remains the treatment of choice for treating obstructive coronary arteries. However, the procedure is hampered by in-stent restenosis (ISR), a phenomenon mainly characterized by local inflammation leading to aggressive SMC proliferation and late neoatherosclerosis [106]. Though recent drug-eluting stents (DES) have reduced ISR incidence, DES is not immune to restenosis. Routine angiographic data after using newer-generation devices demonstrates rates of angiographic restenosis of approximately 5–10% [107]. Current guidelines recommend that patients who develop clinical restenosis after DES implantation be considered for repeated PCI with balloon angioplasty or DESs containing the same drug or an alternative antiproliferative drug [108]. Therefore, new therapeutic interventions are required to target vascular pathologies such as atherosclerosis and restenosis and improve clinical outcomes following PCI. Combining targeted integrin therapy with PCI procedures may prevent the recurrence of stent-induced restenosis. Currently, several approaches are available to target integrins, including monoclonal antibodies, peptide inhibitors, and RGD-mimetic small-molecule inhibitors. However, none of them are commercialized for cardiovascular indications because of limitations. First, preclinical small animal models of neointimal hyperplasia do not mimic clinical settings and could be one of the major contributing factors to the lack of reproducibility of preclinical findings. Second, regulation of integrins is a very complex, dynamic, and quick process [109]. The levels of different integrin heterodimer such as αvβ3, αvβ5, or α5β1, expressed by SMCs may differ during the process of neointima formation, leading to a dynamic change in integrin pattern. By understanding the regional and temporal regulation of integrin expression’s during SMC migration and phenotypic modulation, we may see more success in developing new interventions.

## 10. Conclusions and Future Perspectives

Integrins are cell surface receptors that are involved in mediating cell–cell interaction and cell–ECM interactions. SMCs express several integrins, which are differentially expressed depending on the phenotypic state. Multiple mechanisms regulate integrin bi-directional signaling in SMC, enabling them to proliferate, migrate, and differentiate into synthetic phenotype. Previous preclinical and clinical studies emphasizing αvβ3 and αvβ5 for inhibition of restenosis were met with failures. Nevertheless, understanding the role of other integrins in SMC biology could lead to new interventions to combat restenosis.

## Figures and Tables

**Figure 1 cells-11-00646-f001:**
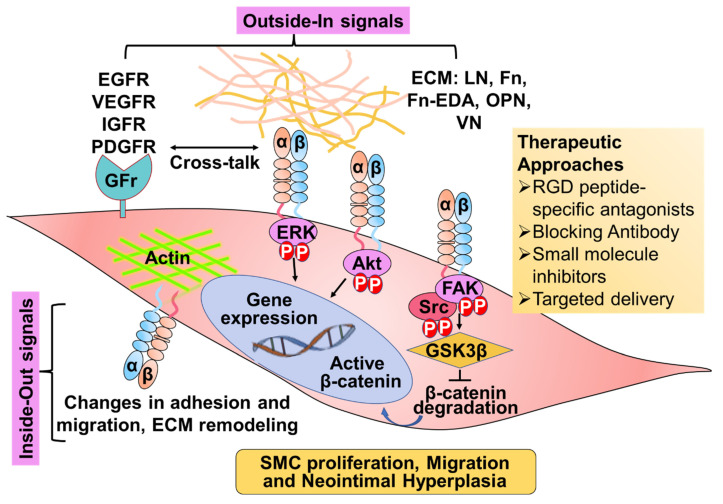
Schematic showing the signal transduction pathways regulated by integrins in smooth muscle cells (SMC). Depending on the type of integrin and its expression on SMCs, they can trigger signals promoting synthetic or paradoxically a contractile SMC phenotype. Many of the reported SMC-specific integrins promote synthetic SMC phenotype. For example, integrin binding to extracellular matrix (ECM) or activation of growth factor receptors (GFr) facilitates downstream signaling events via FAK-Src, Akt, or ERK pathway, resulting in SMC proliferation and migration and neointimal migration hyperplasia. Abbreviations: ERK: extracellular signal-regulated kinase; ECM: extracellular matrix; EDA: extra domain A; FAK: focal adhesion kinase, Fn: Fibronectin; IGFR: insulin-like growth factor receptor; LN: Laminin; OPN: Osteopontin; PDGFR: platelet-derived growth factor receptor; VEGFR: vascular endothelial growth factor receptor; VN: Vitronectin.

**Figure 2 cells-11-00646-f002:**
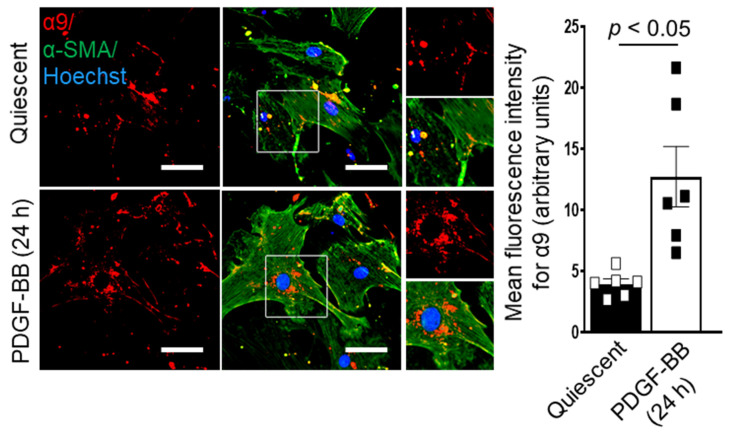
Integrin α9 expression in murine aortic smooth muscle cell (SMC). Serum-starved murine aortic SMC was stimulated with or without platelet-derived growth factor-BB (PDGF-BB) for 24 h. The left panels show representative double immunostaining for α9 (red) and αSMA (green) in SMCs stimulated with or without PDGF-BB. Boxed regions are magnified. Scale bars: 30 μm. The right panel shows the quantification of α9 fluorescence intensity (*n* = 6/group). Statistical analysis: unpaired Student’s *t*-test.

**Table 1 cells-11-00646-t001:** Table representing the expression of different integrin subunits, their implication in smooth muscle cell (SMC) function and disease conditions such as atherosclerosis and neointimal hyperplasia, and integrin-directed drugs used in clinics. Collagen, Col; Laminin, LN; Fibronectin, Fn; Vascular cell adhesion molecule, VCAM; Osteopontin, OPN; Tenascin, TN; Vitronectin, VN; Fibrinogen, Fib; EDA, extra domain A.

Integrin	ECM	SMC Expression	SMC Function	Implication in Atherosclerosis/Restenosis	Integrin-Targeting Agents in Clinics	Reference
α1β1	Col 1-IV, LN	High expression in resting SMCs. Downregulated in culture conditions and during neointimal hyperplasia	Promotes SMC adhesion and contractile phenotype	α1β1 deletion induces a stable plaque phenotype	SAN-300	[10,34,42,52,53,54,55]
α2β1	Col 1 and IV, LN	Undetectable levels in normal human SMCs, and high expression in cultured SMCs	promote chemotaxis of arterial SMCs	α_2_β_1_ deletion had no effect on atherosclerosis	Vatelizu-mab	[16,39,42,52,56]
α3β1	Col 1, Fn, and LN	Detectable levels in normal human SMCs, and high expression in cultured SMCs	No conclusive reports			[10,33]
α4β1	Cellular-Fn, VCAM, OPN	Undetectable levels in normal human SMCs, expressed in SMCs in culture and in intimal atherosclerotic thickening	Induction of SMC differentiation	blocking α_4_β_1_ prevents neointimal hyperplasia	Natalizu-mab AJM300	[38,44,45,52]
α5β1	Fn and LN	Low levels in normal human SMCs, and high expression in cultured SMCs and during neointimal hyperplasia	Promote SMC proliferation and migration	Mediates early atherosclerosis	Volocixi-mab ATN61	[41,57,58]
α7β1	LN	High levels in normal SMCs, and low expression in synthetic SMC	Promotes contractile SMC phenotype	α7 deletion promotes neointimal hyperplasia	No conclusive reports	[20,51,59,60]
α8β1	Fn, TN, VN	Overexpressed in SMCs that display a contractile phenotype low expression in synthetic SMC phenotype and during neointimal hyperplasia	Promotes contractile SMC phenotype. Prevents SMC proliferation and migration	α8 deletion aggravates intimal thickening	No conclusive reports	[12,35,50,61]
α9β1	Fn-EDA, TN, VCAM	Expression increases in synthetic SMC phenotype	Promotes SMC proliferation, migration, and synthetic phenotype.	α9 deletion prevents NH	ASP5094	[21,62]
αvβ1	VN, Fn	Weakly expressed in normal SMCs, and upregulated in SMCs cultured on fibronectin	Inhibits contractility in SMC exposed to serum	No conclusive reports	PLN-74809 PLN-1474	[41,63,64]
αvβ3	VN, OPN, Fn	Weakly expressed in normal SMCs, and upregulated in SMCs cultured on fibronectin and during neointimal hyperplasia	Promotes SMC adhesion, proliferation and migration	Promotes neointimal hyperplasia	LM609, Abcixi-mab (c7E3Fab; ReoPro), Vitaxin, Intetumu-mab, Cillengitide	[16,41,65,66,67,68]
αvβ5	Fib, Fn, OPN VN	highly abundant in cultured SMCs, upregulated upon vascular injury	Promotes SMC adhesion and migration	Promotes neointimal hyperplasia	LM609 Intetumu-mab	[67,69]

## Data Availability

Not applicable.

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
