# Peer review of "Role of Integrins in Modulating Smooth Muscle Cell Plasticity and Vascular Remodeling: From Expression to Therapeutic Implications"

_cells, 2022, doi:10.3390/cells11040646_

Round 1

Reviewer 1 Report

In this manuscript, Jain and Chauhan provided an overview about the role of integrins in vascular smooth muscle cells physiology. The text reads well, the authors used the main references in the field and raised a relevant discussion about integrin-based pharmacological strategies to target vascular diseases such as in-stent restenosis. Please find below the points requiring attention to clarify the text:

  1. In the abstract, the sentence “Smooth muscle cells (SMCs) are the predominant cell types in the blood vessels” should be reformulated, as it gives a wrong notion about the heterogeneity of the vascular wall and the relative contribution of SMCs.
  2. In line 25, it is important to emphasize that vascular tone is seen only at resistant arteries and as a synergic action between vasodilators/ vasoconstrictors and vascular smooth muscle cell contractility.
  3. SMC dedifferentiation is commonly described as a negative phenotype. However, it is required to physiological vascular responses including the replacement of senescent SMCs and the adaptation of vein grafts used as bypasses in cardiac revascularization. Thus, although SMC dedifferentiation and markers such as CD68 and osteopontin (Line 42) may coincide under pathological conditions, these markers do not account for features of dedifferentiated SMCs in general. In addition, it is relevant to observe that contractile/synthetic phenotype coexist, and that instead of a bionomic behavior, there are shades of SMC phenotypes according to physio(patho)logical conditions.
  4. Line 106: Src is short for Sarcoma (reads “sarc”) and should not be confused with steroid receptor coactivator (p160/NCOA).
  5. Lines 178-179: RGD sequences are present on fibronectin molecules and recognized by integrins. The authors stated otherwise.
  6. Did the authors expect the pattern of integrin staining depicted on Figure 2, upper panel? Integrins are usually found at the tips of actin filaments, while the image shows a random pattern. If possible, please provide alternative integrin stains to validate the a9-integrin staining pattern.

Reviewer 2 Report

Critical evaluation of a review article „Role of integrins in modulating smooth muscle cell plasticity and vascular remodeling: from expression to therapeutics implications” by Manish Jain and Anil K. Chauhan, submitted to the special issue of Cells journal- Integrin Activation and Signal Transduction.

Artherosclerosis is a leading cause of morbidity and mortality globally. Vascular pathologies result from abnormal development and behavior of smooth muscle cells (SMCs). We need novel therapies preventing artherosclerosis and also supporting post-surgical intervention treatment. Modern approaches consider molecule-targeted therapies a great chance to fight many diseases such as muscular dystrophies or cancers. Overexpression or downregulation of various integrins has been observed in many cases. Thus, it seems to be justified to study the role of integrins in SMCs. The results of such research have not been reviewed extensively yet.

The authors prepared a well-written, comprehensive review on the role of lectins in SMCs functioning and connection to artherosclerosis and restenosis. Up-to-date results are summarized in Table 1. The authors refer to 9 of their previous papers, all of which refer to the subject of the review. The article content is well-balanced. A short but accurate section introducing lectins family has been prepared. The authors discuss the lectins expression in SMCs and their role in neointimal hyperplasia. A separate section deals with the role of α9- integrin, which has not been studied widely regarding vascular diseases but seems to be much involved in them.

The authors submitted a well-prepared and scientifically sound review.

Author Response

No comments and concerns related to English.

Reviewer 3 Report

1. Integrins are cell surface receptors that are involved in mediating cell-cell interaction 309 and cell-ECM interactions. Furthermore, SMCs express several integrins, can the author point out that the specific site of action in tegrins, which kind of vessels?

2. Thorough editing of the English language and style are still required. Moreover, the logical sequence of the sentences must be improved. In addition, the discussion is sometime confusing, and the authors make some statements not fully supported by the reported research results and experiment It would be great for the author to improve these flaws.

Author Response

1. Integrins are cell surface receptors that are involved in mediating cell-cell interaction and cell-ECM interactions. Furthermore, SMCs express several integrins, can the author point out that the specific site of action integrins, which kind of vessels?
Response: The integrins bind their ligands (components of ECM proteins) via the extracellular domain and play a pivotal role in the development, maintenance, and remodeling of the vasculature. In addition, integrins bind to cytoskeletal proteins of SMCs, including talin, vinculin, α-actinin, and filamin, and play a key role in SMC biology. Many integrins such as αvβ3, α5β1, and α8β1 bind an RGD (arginine-glycine-aspartic acid)-motif present in the ECM protein, including fibronectin. In addition, many non-RGD recognizing integrin such as α4β7 and α9β1 recognize several non-RGD sequences. For example, integrin α9β1 recognize SVVYGLR in OPN, AEIDGIEL in Tenascin-C, and PEDGIHELFP in cellular Fn containing EDA. 

2. Thorough editing of the English language and style are still required. Moreover, the logical sequence of the sentences must be improved. In addition, the discussion is sometime confusing, and the authors make some statements not fully supported by the reported research results and experiment It would be great for the author to improve these flaws.
Response: The manuscript has been read and edited by English speaking Colleague. The logical sequence of some of the sentences were improved. To our best of the knowledge the reported research results are correct and we have extensively revised the manuscript to improve any flaws in discussion.